# Evolution of Microstructure and Mechanical Properties of Al-Zn-Mg-Cu Alloy by Extrusion and Heat Treatment

**Jun Li** [1,†], **Yayun He** [2,†], **Xi Zhao** [2,*] and **Chankyung Kim** [3,*]

1   School of Chemical Engineering and Technology, North University of China, Taiyuan 030051, China; lijun2015@nuc.edu.cn
2   College of Mechatronics Engineering, North University of China, Taiyuan 030051, China; hyy798646088@163.com
3   Department of Chemistry and Chemical Engineering, Inha University, 100 Inha-ro, Michuhol-gu, Inchenon 22212, Korea
*   Correspondence: zhaoxi_1111@163.com (X.Z.); kckyung@inha.ac.kr (C.K.)
†   These authors contributed equally to this work.

**Abstract:** Herein, composite extrusion deformation and heat treatment process at various temperatures were studied on a new type of Al-Zn-Mg-Cu alloy billet. The influence of pre-deformation and the final forming of extrusion and heat treatment of annular channel corner extrusion on the microstructure evolution and mechanical properties were explored. The results show that the extrusion process could further refine the structure and break the coarse θ phase. The grains can be refined again after the deformed sample was treated by solution-aging treatment. At the same time, a fine, dispersed second phase is precipitated around the fine recrystallized grains. This is the main reason for the increase in alloy elongation and tensile strength. The best heat treatment process parameters for the formed cup-shaped structure are 480 °C × 1 h solid solution and 120 °C × 24 h aging. The strengthening of the alloy mainly includes three mechanisms: fine grain strengthening, precipitation strengthening, and dispersion strengthening.

**Keywords:** new Al-Zn-Mg-Cu alloy; annular channel corner extrusion; heat treatment; strengthening mechanisms of alloy

## 1. Introduction

Lightweight is an important development direction in the fields of aerospace and national defense. Aluminum alloy has become an ideal structural material in the aerospace field by its easy processing, lightweight, corrosion resistance, and high strength and toughness [1]. Cup-shaped components are important basic load-bearing components in the aerospace, so how to manufacture high-performance aluminum alloy cabin components has become the focus and difficulty of research in the aerospace field. Although traditional back extrusion has the advantages of simple mold and easy operation, this method of deformation has large forming force, small plastic deformation, and uneven deformation, so this method is not suitable [2,3]. The annular channel corner extrusion process can use small-diameter billets to form large-diameter cabin shells at one time, which can shorten the process flow and improve production efficiency. The shear deformation applied by the corners of the mold obtains a large amount of plastic deformation, which can be significantly thinner. The crystal grains are reduced, and the overall mechanical properties of the components are greatly improved [4–7]. Zhao et al. [8,9] used AZ80 alloy to compare the traditional reverse extrusion and annular channel angular extrusion process. They found that the annular channel angular extrusion process can better refine the grain structure and make the structure more uniform and at the same time obtain better tensile properties. The billet needs to be prefabricated before the component is prepared. To keep the shape of the billet unchanged after pre-deformation, pier extrusion is selected as the pre-deformation



process of the billet. Therefore, the final forming method of cabin components is cyclic pier extrusion and annular channel angular extrusion.

The new Al-Zn-Mg-Cu alloy is a heat-treatable aluminum alloy. The heat treatment process of aluminum alloy components can improve the alloy structure and performance and meet the requirements of use [10–14]. The structure of the metal will change to a certain extent after plastic deformation and will have a certain impact on the subsequent heat-treatment process. Therefore, this paper discusses the influence of the forming process on the alloy structure and properties of the formed aluminum alloy components, studies the heat treatment process, and explores the relationship between the structure and properties formed by the heat treatment. A theoretical basis for the popularization and application of the new Al-Zn-Mg-Cu alloy was provided.

## 2. Materials and Methods

In this experiment, a new type of Al-Zn-Mg-Cu alloy homogenization cast rod provided by Central South University (Changsha, China) was used, and the homogenization parameter was 470 °C × 20 h. The specific chemical composition is shown in Table 1.

**Table 1.** Aluminum alloy composition (%, mass fraction).

| Element | Al | Zn | Mg | Cu | Zr | Fe | Si | Ti |
|---|---|---|---|---|---|---|---|---|
| Content | 86.0–88.9 | 8.0–9.3 | 1.5–2.4 | 1.2~1.9 | 0.15 | 0.06 | 0.03 | 0.15 |

Firstly, the billet was prefabricated by extrusion process, and then the cylindrical component was prepared by the annular channel angular extrusion process. The forming process is shown in Figure 1, and the formed part is shown in Figure 2. First, wire cutting was used to take the initial billet with a size of Φ140 mm × 200 mm, as shown in Figure 2a. The billet was preheated at 470 °C for 3 h. The preheating temperature of the die is 20 °C higher than that of the billet, and the preheating is 3.5 h. The extrusion pre-deformation experiment was carried out by a 630-ton hydraulic press, as shown in Figure 1a, and the billet was upsetting. After that, the billet was extruded as shown in Figure 1b, and the diameter of the billet changed from 140 to 100 mm. The upper and lower ends of the pre-deformed billet were processed into planes and processed into bars of 90 mm × 360 mm, as shown in Figure 2b. Before the angular extrusion of the annular channel, the pre-deformed billet was kept at 430 °C for 2.5 h, and the preheating temperature of the die is 20 °C higher than that of the billet for 3.5 h. After putting the heat-preserved billet into the mold, the annular channel angular extrusion test was carried out by a 630-ton hydraulic press. Figure 1c,d show the beginning and ending stages of annular channel angular extrusion, respectively. The extruded formed parts and the section of formed parts are shown in Figure 2c,d, respectively. The thickness of the cup wall after extrusion is 20 mm, the inner diameter of the cup wall is 160 mm, and the height is 220 mm.

The middle area of the cup wall was taken as a sample to study the heat treatment process. The sample size was 10 mm × 10 mm × 10 mm. According to the DSC (Differential Scanning Calorimetry) test and analysis results, the aging process was formulated, and the specific heat treatment plan is shown in Table 2.

We grind the sample surface with 300#, 500#, 1000#, 2000#, 3000#, 5000#, and 7000# in turn, then perform mechanical polishing, and perform Vickers Hardness (HV) Test on the polished sample. The HV pressure, indenter speed, and pressure holding time are 200 gf, 25 μm/s, and 15 s, respectively. To ensure the accuracy of the HV results, 12 different positions were selected for dispersion. The average value was taken as the hardness value after eliminating maximum and minimum values. The surface of the polished sample was treated with the EMRES102 (Leica, Berlin, Germany) ion thinner with an ion beam voltage of 6.7 kV and a current of 2.7 mA. A scanning electron microscope (SEM, Hitachi SU5000, Tokyo, Japan) was used to characterize the structure, including the morphology, distribution, and size of the second phase. The element content was

determined using energy dispersive spectroscopy (EDS, Hitachi SU5000, Tokyo, Japan), and the sample structure was analyzed and processed using electron backscatter diffraction (EBSD Mahwah, NJ, USA) and EDAX-OIM v7.3 (Post-processing analysis) software.

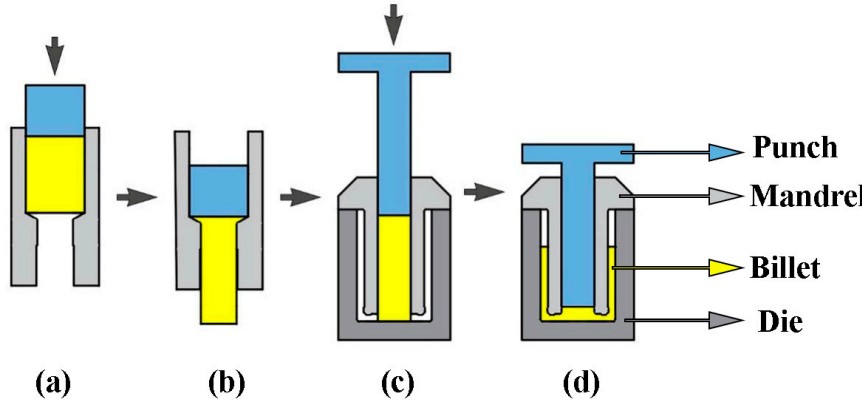

**Figure 1.** Diagram of forming process (**a**) upsetting, (**b**) extrusion, and (**c,d**) the beginning and ending stage of annular channel angular extrusion.

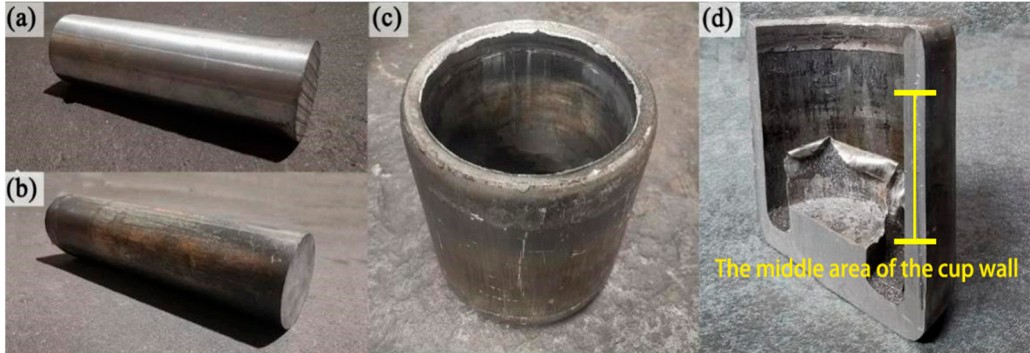

**Figure 2.** Schematic diagram of the formed part. (**a**) initial billet, (**b**) after upsetting and extrusion billet, (**c**) formed parts, and (**d**) the section of formed parts.

**Table 2.** The plan of the heat treatment process.

| Solution Temperature (°C) | Solution Time (h) | Aging Temperature (°C) | Aging Time (h) |
|---|---|---|---|
| 480 | 1 | 120, 130, 145, 160 | 0–60 |

The solid solution model of Al-Zn-Mg-Cu alloy was constructed according to the specific situation of the experiment as shown in Figure 3. Firstly, the unit cell of the alloy was built by referring to the composition of each metal in Table 1, i.e., Zn: 9.3%, Mg: 2.4, Cu: 1.9, and Al: 86.4%. Then, the solid alloy model was built by extending the unit cell and was simulated by molecular dynamics (MD) with the help of the Forcite module in Materials Studio (MS) software [15,16]. COMPASS (Condensed-phase Optimized Molecular Potentials for Atomistic Simulation Studies) was selected because Al, Zn, Mg, and Cu metals can be properly described in this force field [17,18]. NVT (Constant volume/constant temperature dynamics) ensemble was used to simulate for a short time to balance the pressure of the system. Velocity Scale thermostat, which can quickly correct kinetic energy, was used. NPT (Constant pressure/constant temperature dynamics) ensemble was used for MD simulation of the system after the system was balanced, and the total simulation time was 500 ps. The NHL (Nosé-Hoover-Langevin (NHL) thermostat is a method for performing constant-temperature dynamics) thermostat [19] and Andersen barostat were used to control temperature and pressure, respectively. Next, the Al-Zn-Mg-Cu liquid alloy

model was built using the Amorphous Cell module. The size of the liquid alloy model was consistent with that of the solid alloy model. MD simulations of NVT and NPT were also performed for the liquid alloy model using the same method applied to the solid model, to confirm the density of liquid alloy. Next, an equal amount of liquid alloy was added on top of the solid alloy. Then MD simulations were performed at different temperatures, and the parameter settings were the same as before. The temperature setting was the same as that of heat treatment, as shown in Table 2. Finally, the mechanical properties of the Al-Zn-Mg-Cu alloy model at different temperatures were calculated to study the effect of heat treatment on the alloy.

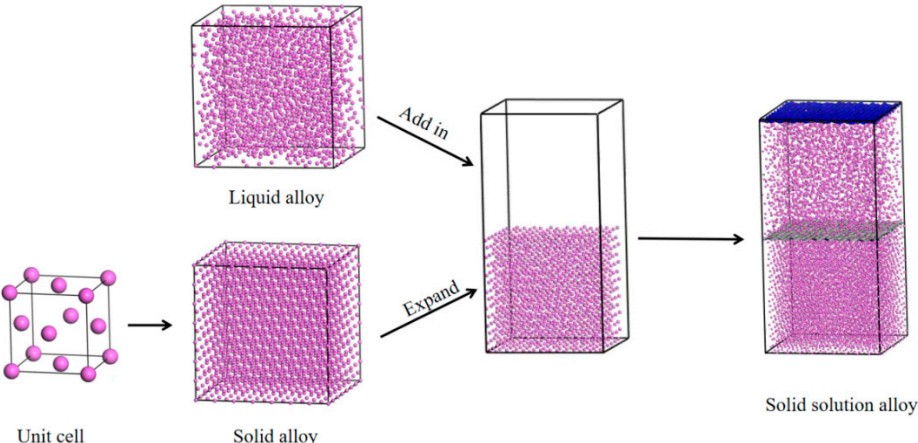

**Figure 3.** The solid solution model of the Al-Zn-Mg-Cu alloy construction process.

Tensile test is used to measure the axial load of the specimen. Mechanical indexes such as tensile strength, yield strength, and elongation can be measured with an extensometer. The equipment used is Instron 3382 (Instron, Boston, MA, USA) universal tensile testing machine. In this paper, the tensile rate is unified as 1 mm/min, the lower end of the stretching rod is clamped with the chuck, and the upper end of the stretching rod is clamped after the load was adjusted to zero. Take three tensile rods in the same state were taken and their average value was taken as the test result. In this paper, the stretching rod sampling always ensures that the extension rod axis is parallel to the extrusion direction.

Vickers Hardness was used for hardness measurement in this paper. The specific parameters were as follows: pressure 200 gf, pressing speed 25 μm/s, and holding time 15 s. The polished sample was measured 12 times at different positions of the sample, and the average value was taken after removing the two maximum values.

The sample preparation process is as follows: $7 \times 7 \times 7$ samples with flat upper and lower surfaces are cut by the all-in-one finishing machine; The samples were roughed, polished, roughed and polished in turn until no obvious polishing marks were found 200 times under the optical microscope. After confirming that there is no wear and debris throwing, put it into the ion thinning instrument and thin the test surface. To prevent oxidative contamination of the sample surface, the prepared sample should be tested as soon as possible.

The observation test process is as follows: stick the sample in the center of the sample table with a 70° tilt with conductive adhesive, push it into the experimental chamber and vacuum; The scanning position (generally center) was determined under the electron microscope, and the working distance was adjusted to 15mm. According to the grain size and distribution, the image magnification and focal length were adjusted until the sample morphology was clear. Map in EBSD system, determine scan step size according to need, and finally collect spectrum to complete the test. It is generally considered that the orientation of a grain is best determined by 3–5 step sizes. According to the metallographic image, choose 200–400 magnification, scanning step size is 0.5−0.6 μm. Each scan takes about 4–5 h. OIM analysis software was used to analyze EBSD data.

## 3. Results and Discussion

### 3.1. Influence of Extrusion on the Microstructure of the Alloy

The homogenized state and deformed morphology structure are shown in Figure 4. From Figure 4a, it can be observed that the original grain structure of the homogenized state is relatively coarse and equiaxed, and the average diameter of the crystal grains is about 400 μm. Figure 4b shows that the alloy is affected by the extrusion force during the pre-deformation. The grains appear to have a fibrous distribution along the extrusion direction—the grain boundaries and grains. A fine recrystallized structure was produced inside, discontinuous large-angle grain boundaries were formed in the middle of some grains, and the average grain size of the alloy was reduced to 200 μm. This is because, in the process of extrusion deformation, the dislocation cross-slip and climbing occurred with the continuous deformation. The cellular substructure in the grain gradually absorbed the dislocations accumulated on the subgrain boundary, the orientation difference of the adjacent subgrain increases gradually, and the subgrain boundary was transformed into a large-angle grain boundary, which promoted the generation of continuous dynamic recrystallization. Figure 4c shows the structure after annular channel angular extrusion. In this structure, the grain boundary of elongated fibrous grains was flatter, the grain width was further reduced, and the size of some recrystallized grains grew. According to statistical analysis, the proportion of recrystallized grains was increased, and the average grain size was decreased to 132 μm. This was due to the continuous production of new fine dynamic recrystallized grains. To reduce the surface energy and keep the recrystallized grains in a stable and low free energy state, the newly recrystallized grains were absorbed by the migration movement of the grain boundaries, and the recrystallized grains grew up. More recrystallized grains were produced in the fibrous deformation grains and at grain boundaries.

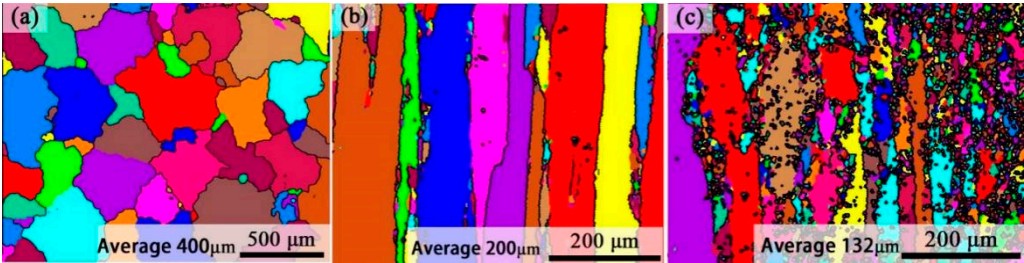

**Figure 4.** EBSD diagram of alloy microstructure. (**a**) original grain structure. (**b**) pre-deformed structure. (**c**) cup-shaped structure after annual channel angular extrusion.

The sample orientation difference is calculated after pre-deformation and annular channel angular extrusion deformation. As shown in Figure 5, the proportion of misorientation angle greater than 45° is higher for cup-shaped component than pre-deformation one. The results show that the recrystallization occurs further after the angular extrusion of the annular channel, which further increases the orientation difference in the sample and the degree of dynamic recrystallization.

The proportion of large misorientation angle of the cup-shaped structure is higher than that of the pre-deformed one. This is because the forming temperature of the cup-shaped component is lower than that of the pre-deformation structure, which strengthens the work-hardening effect to a certain extent. Therefore, in the grain orientation difference diagram, the proportion of smaller and larger misorientation angle is increased.

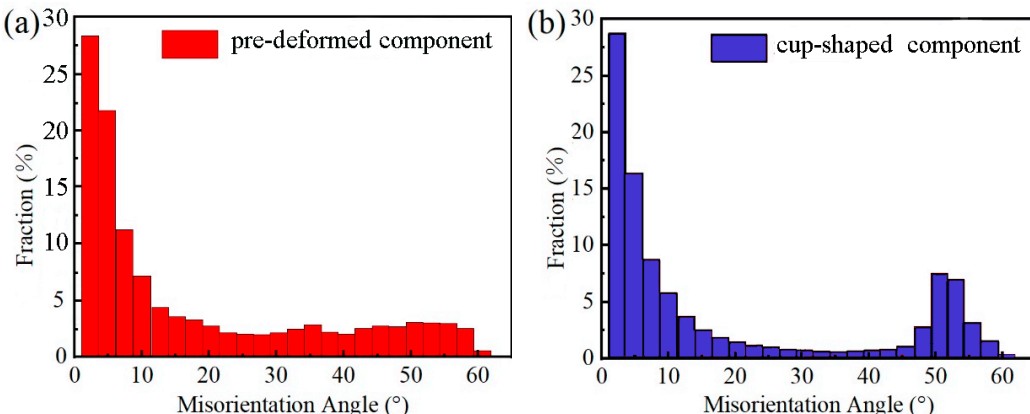

**Figure 5.** Distribution map of misorientation angle of the deformed structures. (**a**) pre-deformed component. (**b**) cup-shaped component.

　　SEM pictures of the microstructures of sample materials are shown in Figure 6. The homogenized scanning structure, Figure 6a, shows some coarse network phases in the sample. Scanning specific points through EDS can measure the element distribution ratio. The atomic ratio of Al, Zn, Mg, and Cu is 59.2:6.76:5.85:28.2. The atomic ratio of Al to Cu is close to 1:2. Therefore, the composition at this point is $CuAl_2$, which is the θ phase. Small rod-like phases are distributed on its edge. Figure 6b is the scanned structure of the sample after pre-deformation. It can be observed that the coarse θ phases in the network are broken because of extrusion deformation, distributed along the extrusion direction, and the θ phases are no longer observed in the network structure. Figure 6c shows that with the progress of the angular extrusion deformation of the annular channel, these coarse θ phases are further fragmented. The size is further reduced, and they are more evenly distributed in various places. Compared with Figure 6a,b, the fine second phase in Figure 6c has a higher degree of dispersion. These dispersed second phases can effectively pin the movement of grain boundaries, hinder the growth of recrystallization, and achieve the effect of grain refinement.

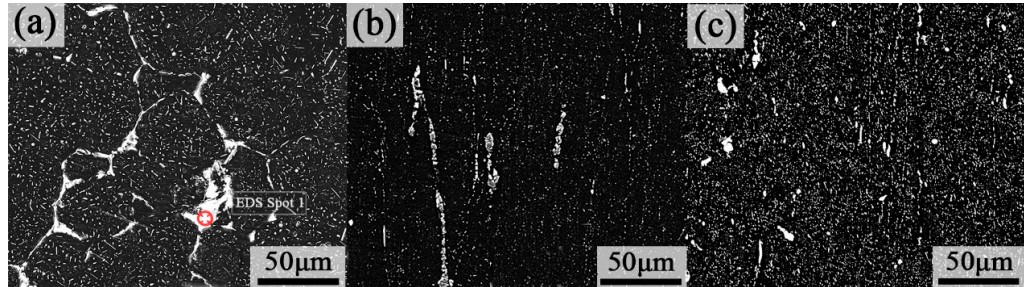

**Figure 6.** Backscattered electron micrographs (**a**) homogenization state, (**b**) after pre-deformation, and (**c**) after angular extrusion deformation of the annular channel.

### 3.2. Effect of Heat Treatment on Alloy Structure and Properties

　　The solid solution and aging heat treatment process (T6) of the formed structures were studied. The solution process was done at 480 °C for 1 h, and the aging temperature was determined according to the DSC curve. As shown in Figure 7, there is an obvious endothermic peak at 123.1 °C in the DSC curve. This peak is the dissolution peak in the GP zone, and the exothermic peak at 165.7 is the η′-phase precipitation peak. The η′-phase is the main strengthening phase of the Al-Zn-Mg-Cu alloy, so the aging treatments were performed at 120, 130, 145, and 160 °C with different aging time scales.

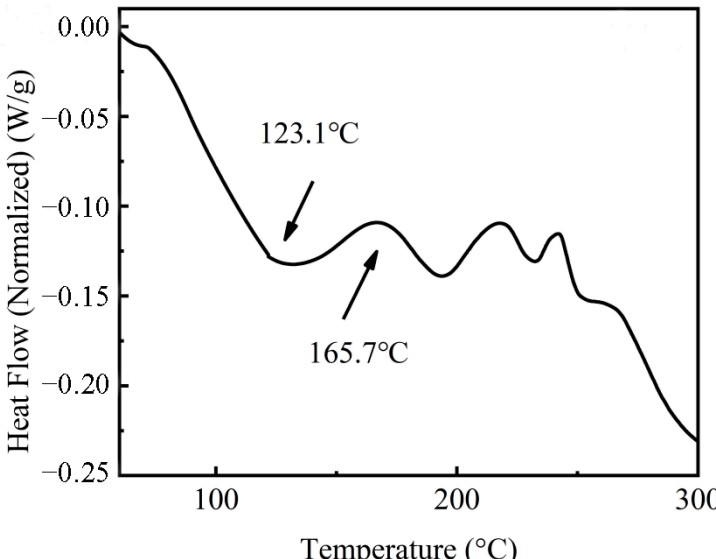

**Figure 7.** DSC curve for the solid solution.

The aging experiments for the solid solution samples were divided into four sets. The first set is aging at 120 °C, five times every 12 h. The second set is aging at 130 °C, six times every 4 h. The third set is aging at 145 °C, six times every 3 h. The fourth set is aging at 160 °C, 6 times every 2 h. The Vickers Hardness values of the heat-treated samples were measured, as shown in Figure 8. This figure shows that changes in the HV values exhibit a similar trend under different aging temperatures. In proportion to the aging time, the hardness increases rapidly at the initial stage and then gradually decreases after reaching the peak value. This is because as the aging progresses, the precipitation ratio of the second phase gradually increases to enhance the precipitation strengthening, making the alloy hardness reach the peak. When the aging time increases further, the solute elements Mg and Zn in the matrix are gradually consumed, and the precipitated phases grow to a certain extent, which reduces the degree of dispersion between the precipitated phases and weakens the precipitation strengthening effect, resulting in a decrease in the hardness of the alloy. Comparison of the time to reach the peak at different temperatures reveals that the higher the aging temperature, the shorter the time for the hardness to reach the peak. For the four experimental sets, the times to reach the peak hardness are 24, 12, 9, and 4 h and the corresponding hardness values are 211.6, 202.8, 203.8, and 202.6 HV for the sets 1, 2, 3, and 4, respectively. This is due to the increase of the aging temperature, which accelerates the precipitation behavior of the second phase and makes the sample reach the peak hardness earlier. As the highest aging hardness of 211.6 HV is achieved at 120 °C × 24 h, this condition is selected as the best aging parameter, which is greatly improved compared with the 75 HV hardness value of the cup-shaped sample.

The model of Al-Zn-Mg-Cu solid solution alloy was simulated by the MD method at four different temperatures, and its mechanical property parameters were calculated, as summarized in Table 3. E, G, and K are the tensile modulus, shear modulus, and bulk modulus of the material, respectively. K/G value is used to evaluate the toughness of the material. The larger the value, the better the toughness of the material. It can be seen from Table 3 that with the increase of temperature, the toughness of the Al-Zn-Mg-Cu solid solution alloy first increases and then decreases. The reason is that with the initial increase of temperature, the precipitation of the second phase accelerates, which increases the toughness of the material. But, with the continuous increase of the temperature, the precipitated phase will weaken the precipitation effect accompanied by the decrease in toughness, which is consistent with the previous finding.

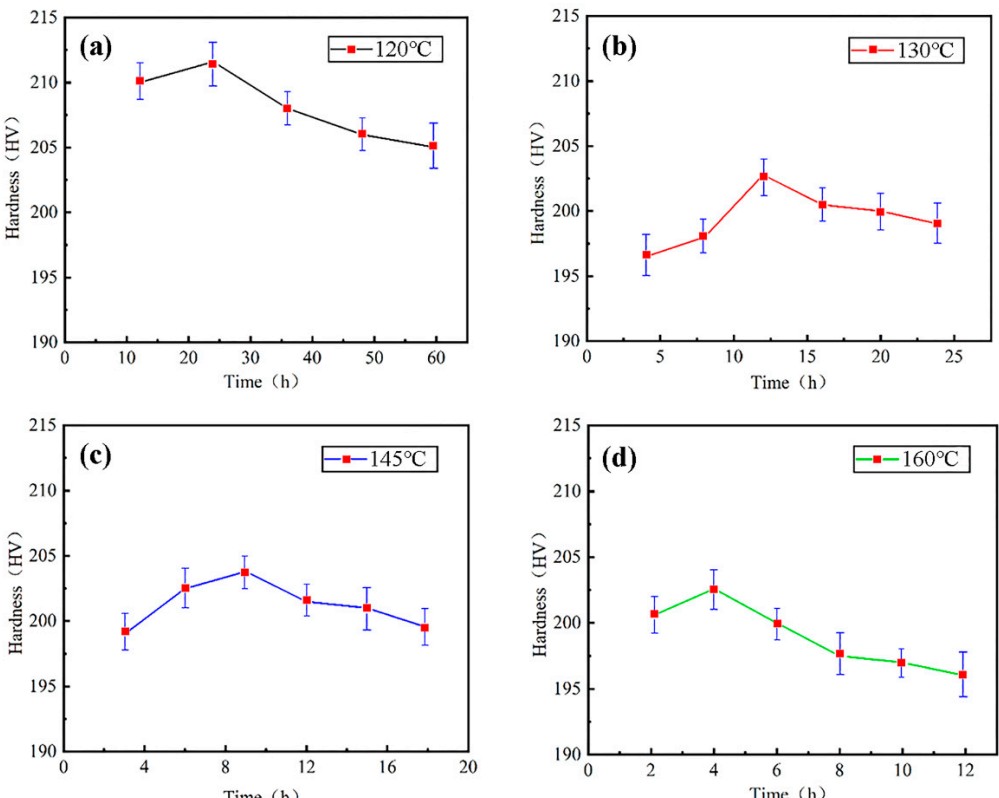

**Figure 8.** Aging hardness curve under different temperatures (**a**) 120 °C, (**b**) 130 °C, (**c**) 145 °C, and (**d**) 160 °C.

**Table 3.** Mechanical property parameters of Al-Zn-Mg-Cu solid solution alloy at different heat treatment temperatures.

| Temperature (°C) | E | G | K | K/G |
|---|---|---|---|---|
| 120 | 80.35 | 49.96 | 19.24 | 0.39 |
| 130 | 81.80 | 47.83 | 21.14 | 0.44 |
| 145 | 78.92 | 50.71 | 18.22 | 0.36 |
| 160 | 79.07 | 51.06 | 18.15 | 0.36 |

The scanning organization of the peak-aged samples is observed from the experiment at different aging temperatures. It can be seen from Figure 9 that, under different temperature conditions, fine precipitated phases are precipitated after heat treatment, and the precipitated phases are in a state of dispersion, mainly distributed at the grain boundary, which shows that the grain boundary of the crystal grain provides a favorable position for the precipitation of the phase. These fine dispersed second phases can effectively pin the dislocations, hinder the movement of the dislocations, and achieve the effect of strengthening the properties of the alloy. After the aging process, some larger size of precipitated phases (pointed by the arrow) appears and the lengths of the η′-phase shown in Figure 9a–d are 0.43, 0.83, 1.21, and 1 μm, respectively. It is found that as the aging temperature increases, the size of the precipitated phase increases, which will reduce the degree of dispersion of the precipitated phase to a certain extent.

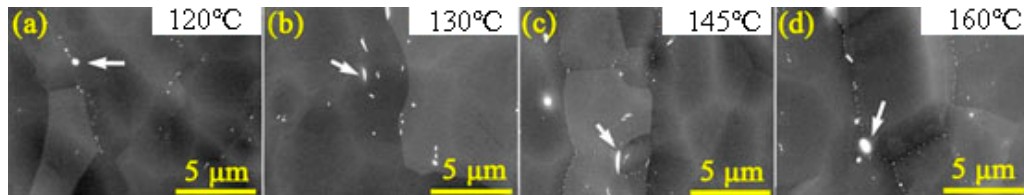

**Figure 9.** SEM images of Peak aging microstructure at different aging temperatures. (**a**) 120 °C; (**b**) 130 °C; (**c**) 145 °C; (**d**) 160 °C.

Close examination of the microstructure of the cup-shaped structure under the best aging parameters (see Figure 10) shows that a large amount of static recrystallization occurred after the T6 process treatment of the formed structure, and the average grain size of the static recrystallization is 6 μm. It can be seen from Figure 5b that the dislocation density inside the crystal grains of the cup-shaped piece is relatively high, so it has a strong driving force [20]. Under the thermal activation of the heat treatment, the unlike dislocation in the alloy is eliminated, the dislocation density is reduced, and the dislocation cell walls with small misorientation form sub-grain boundaries [21]. The dislocation cells merge into sub-grains and transform into grain boundaries, resulting in many static recrystallized grains. The proportion of recrystallized grains in the alloy is greatly increased, and the overall average grain size is reduced to 112 μm.

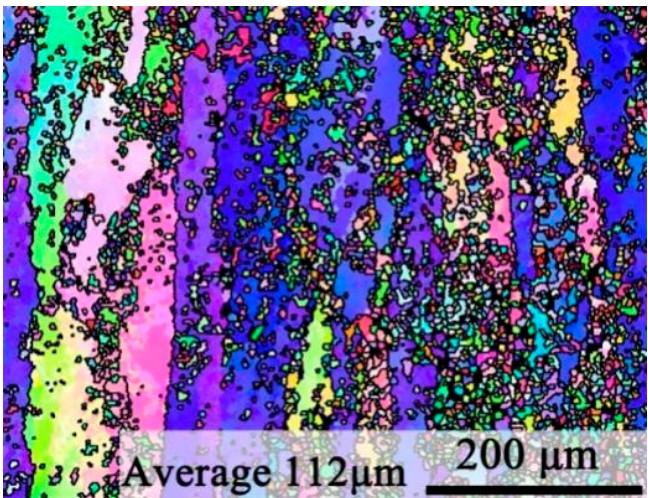

**Figure 10.** Microstructure after heat treatment (480 °C × 1 h solid solution and 120 °C × 24 h aging).

Figure 11 shows the plots of the mechanical properties of the alloy under the original homogenization state and after the optimal heat treatment. When the initial homogenized billet is heat-treated, the elongation of the alloy changes from 4% to 15.7%, and the tensile strength changes from 289 to 630 MPa, which suggests that the material obtained better tensile strength and elongation. Grain refinement enables the material to disperse the force on more crystal grains when plastic deformation occurs so that the plastic deformation is uniform, the stress concentration is reduced, and the metal obtains a better deformation coordination ability, thereby improving the elongation of the material. According to Hall-Petch formula theory [22], grain refinement can improve alloy strength. After heat treatment, the fine and dispersed precipitated phase greatly improves the strength performance of the alloy material. Therefore, the alloy strengthening mainly includes three mechanisms: fine-grain strengthening, precipitation strengthening, and dispersion strengthening.

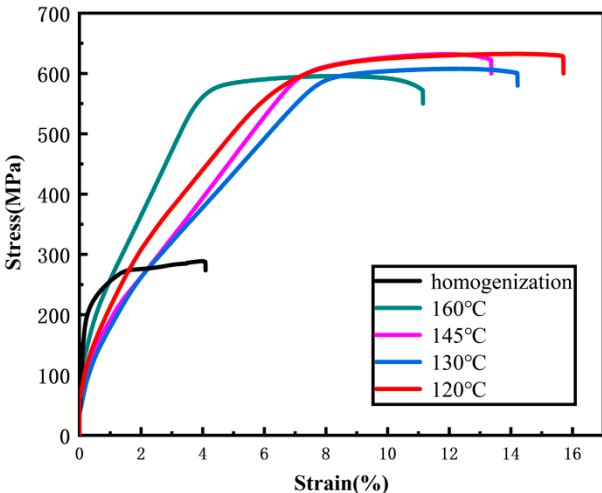

**Figure 11.** Mechanical properties of the alloy under the homogenization state and heat treatment parameters of the Cup-shaped component.

## 4. Conclusions

(1)　A new type of Al-Zn-Mg-Cu alloy cup-shaped component is prepared by high- and low-temperature composite extrusion deformation, and continuous dynamic recrystallization. The cooling ring channel corner extrusion process has stronger grain refinement and crushing θ coarse phase effects, and the average grain size reaches 180 μm after deformation.

(2)　The corner extrusion process enables the sample to obtain a higher dislocation density, which provides a strong driving force for the static recrystallization and the precipitation of the second phase after aging. The size of the aging precipitates increases with the increase of temperature.

(3)　In order to maximize the strength and toughness of the new Al-Zn-Mg-Cu alloy cup-shaped components, the best heat treatment process is 480 °C × 1 h solid solution and 120 °C × 24 h aging. The tensile strength reached 630 MPa after heat treatment, and the elongation reached 15.7%. The strengthening mechanisms of the alloy mainly include fine-grain strengthening, precipitation strengthening, and dispersion strengthening.

**Author Contributions:** Investigation, C.K.; Resources, X.Z.; Writing—original draft, Y.H.; Writing—review & editing, J.L. All authors have read and agreed to the published version of the manuscript.

**Funding:** This research was funded by the Shanxi Scholarship Council of China, grant number 2020-107, Natural Science Foundation of Shanxi Province, grant number 201901D111129, and Inha University Research Fund.

**Institutional Review Board Statement:** Not applicable.

**Informed Consent Statement:** Not applicable.

**Data Availability Statement:** Not applicable.

**Conflicts of Interest:** The authors declare no conflict of interest.

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
