# Peer review of "Evolution of Microstructure and Mechanical Properties of Al-Zn-Mg-Cu Alloy by Extrusion and Heat Treatment"

_coatings, doi:10.3390/coatings12060787_

Round 1

Reviewer 1 Report

The assessed work describes the influence of annealing of the Al-Zn-Mg-Cu alloy on the microstructure and strength properties.

The abstract, introduction and research methodology are presented properly.

The authors indicated the potential application of the obtained material.

The developed research results are properly analyzed and compared with other works on a similar subject.

The summary and the literature cited are selected appropriately.

Should be corrected:

Line 226: change K to degrees Celsius

Temperatures obtained by the DSC method should be given to whole units.

My assessment of the work is high, in general I have no substantive comments on the presented text.

Author Response

Response to reviewers’ comments

Thank you to the reviewers for their time and thoughtful comments, many of which have been incorporated into the revised manuscript. We have read the reviewer’s and editor’s comments very carefully and have extensively revised the manuscript carefully. To make the reviewer comments available to all of the authors, we have repeated the comments from each of the reviewers below (in a different font). The revised portions were marked with revision mode in the revised manuscript. The page and line numbers refer to our revised manuscript submitted 4/5/2022.

Reviewer:

The assessed work describes the influence of annealing of the Al-Zn-Mg-Cu alloy on the microstructure and strength properties. The abstract, introduction and research methodology are presented properly. The authors indicated the potential application of the obtained material. The developed research results are properly analyzed and compared with other works on a similar subject. The summary and the literature cited are selected appropriately. Should be corrected:

Line 226: change K to degrees Celsius

Temperatures obtained by the DSC method should be given to whole units.

Reply:

Thank you for your advice. We have changed K to degrees Celsius.

(Line 226)

Reviewer 2 Report

Review of the manuscript “Evolution of Microstructure and Mechanical Properties of Al-Zn-Mg-Cu Alloy by Extrusion and Heat Treatment”

It is about the influence of pre-deformation and the final forming of extrusion and heat treatment of annular channel corner extrusion on the microstructure evolution and mechanical properties.

The manuscript is interesting, however not completely well organized and discussed. English acceptable.

It can be reconsidered after the following major revisions:

Line 207: use the same number of significant digit.

Dislocation density is introduced and qualitatively discussed in line 287, 289 and in conclusion (line 319). However there is no experimental evidence of that (XRD?). Please add and discuss.

Mechanical properties are described in term of hardness, tensile strength and elongation. However experimental results regard only hardness! Please add further experimental results and discuss!

Furthermore tensile strength and elongation are usually inversely proportional. Please explain why on line 323 tensile strength and elongation reached a maximum value, as described.

Discussion (point n. 3): please add motivation for “the best heat treatment”…

Round 2

Reviewer 2 Report

The manuscript has been improved.

It can be accepted in the present form.